Association between rs7517847 and rs2201841 polymorphisms in IL-23 receptor gene and risk of ankylosing spondylitis: a meta-analysis

Xu Bin 1 2
Ma Jian-xiong 1
Ma Xin-long 1 pubstory@163.com
Jia Hao-bo 1
Feng Rui 1
Xu Li-yan 1
1 Biomechanics Laboratory of Orthopaedic Institute of Tianjin Hospital , Tianjin , China
2 Tianjin Medical University , Tianjin , China
Pintilie Melania
Electronic publication date: 2015 Apr 23
Publication date: 2015
Volume: 3
Electronic Location ID: e910
Received 2014 Dec 14; Accepted 2015 Apr 3
Copyright: © 2015 Xu et al.
Copyright year: 2015
Copyright holder: Xu et al.
License: This is an open access article distributed under the terms of the Creative Commons Attribution License, which permits unrestricted use, distribution, reproduction and adaptation in any medium and for any purpose provided that it is properly attributed. For attribution, the original author(s), title, publication source (PeerJ) and either DOI or URL of the article must be cited.
License URL: https://creativecommons.org/licenses/by/4.0/

Keywords: Ankylosing spondylitis, SNPs, Meta-analysis, Interleukin-23 receptor

Funding: The authors declare there was no funding for this work.

==============================
To comprehensively evaluate the association between rs7517847 and rs2201841 polymorphisms in the Interleukin-23 (IL-23) receptor gene and ankylosing spondylitis (AS), a meta-analysis was performed. The Pubmed, Embase, MEDLINE, Cochrane, China National Knowledge Infrastructure (CNKI), VIP, Wanfang and China Biology Medicine disc (CBMdisc) databases were searched to identify eligible studies on rs7517847 and rs2201841 polymorphisms in the IL-23 receptor gene and AS that were published through September 2014. Data of interest were extracted from each study, and the meta-analysis was performed using STATA 12.0. Four studies were eligible for the meta-analysis and included a total patient population of 2,465. With regards to rs7517847, the current study showed that the genotype GG and allele G might play a protective role during AS (OR = 0.76, 95% CI [0.59–0.99]; OR = 0.88, 95% CI [0.78–0.99] for homozygote and allelic models, respectively). However, according to the meta-analysis, there was no statistical association between the genotype or allele of rs2201841 and an individual’s susceptibility to AS in all genetic models. In conclusion, it was the IL-23 rs7517847 polymorphism rather than the rs2201841 polymorphism that had a statistical association with AS. Nevertheless, more evidence is needed to confirm this result. Consequently, it is necessary to carry out more high-quality studies to confirm the associations between these two single nucleotide polymorphisms and AS.

Introduction

Ankylosing spondylitis (AS) is an autoimmune disease that features chronic inflammation and ossification of the sacroiliac articulation and enthesis of the tendon. Approximately 0.2% of the general population suffers from AS, and its incidence is 0.1–1.4%, 0.2–0.54%, 0.86%, and 2% in European, Chinese, Caucasian, and HLA-B27-positive populations, respectively (Braun & Sieper, 2007; Zeng et al., 2008; Braun et al., 1998; Goldman & Schafer, 2012). Moreover, when one twin suffers from AS, the probability that the other is also an AS patient is greater than 90% (Brown et al., 1997). An obvious gender tendency has been found in AS, i.e., men are more susceptible to the disease compared with women (Goldman & Schafer, 2012).

The etiology and mechanisms behind AS remain unclear. Scientists have thought that susceptibility to AS is strongly affected by heredity. Although HLA-B27 is a confirmed genetic risk factor, not all AS cases can be explained by genetic association, which implies that other factors are also important (Vegvari et al., 2009). Wang et al. (2010) found that there was a significantly higher mRNA expression of the interleukin-23 receptor (IL-23R) gene in peripheral blood monocytes of AS patients compared with those of normal controls. When IL-23 was overexpressed, CD4- and CD8- T cells were affected (Sherlock et al., 2012), which led to peripheral enthesitis and the formation of new bone (Miceli-Richard, 2004). Another study showed that the IL-17/IL-23 axis could be a potential target for AS treatment (Di Meglio et al., 2013). In accordance with a genome-wide association study (GWAS), it was also shown that there was an association between IL-23R and AS in Caucasian patients (Burton et al., 2007).

Thus far, several meta-analyses have investigated the relationship between several single nucleotide polymorphisms (SNPs) and AS; they have found that several IL-23R SNPs were associated with AS in Caucasian populations but that others were unrelated to AS in Asian patients (Karaderi et al., 2009; Chen et al., 2012; Duan et al., 2012; Pimentel-Santos et al., 2009). No meta-analysis has yet examined the relationship between SNPs rs7517847 and rs2201841 and AS. The association between these two SNPs and AS was borderline significant only (Sáfrány et al., 2009; Dong et al., 2013). These negative results may have been caused by a small included population. The current study uses meta-analysis to determine the association between rs7517847 and rs2201841 in IL-23R SNPs and AS for the first time. This study provides more comprehensive evidence for rheumatologists considering the association between IL-23R SNPs and AS.

Materials and Methods

Search strategy and selection criteria

A comprehensive search of databases such as Pubmed, Embase, Medline, Cochrane, China National Knowledge Infrastructure (CNKI), VIP, Wanfang and China Biology Medicine disc (CBMdisc) databases was conducted. Searches included literature dated from database origin to September 2014, and the following key words were used: “IL-23” OR “interleukin-23”, “Ankylosing Spondylitis” OR “AS”, and “polymorphism” OR “polymorphisms”. In the CNKI, VIP, Wanfang and CBMdisc databases we searched for corresponding words in Chinese characters. The full search strategy for the Embase database is presented in Table 1. No language restrictions were used. A manual search for references beyond those in the above-mentioned databases was also implemented. For studies that did not describe genetic distribution data in detail, email correspondence with the main authors was used to complete the data. Titles and abstracts were independently screened by two authors to identify potentially related studies. Full-text versions of the identified studies were reviewed to select those that met the eligibility criteria. The identified studies were subjected to a final confirmation before inclusion in the meta-analysis.

Table 1 The full search strategy for Embase.

#1	‘interleukin’/exp OR interleukin AND 23	
#2	il AND 23	
#3	#1 OR #2	
#4	ankylosing AND (‘spondylitis’/exp OR spondylitis)	
#5	ankylopoietic AND (‘spondylarthritis’/exp OR spondylarthritis)	
#6	ankylopoietic AND (‘spondylitis’/exp OR spondylitis)	
#7	ankylosing AND (‘spine’/exp OR spine)	
#8	ankylosing AND spondylitis	
#9	Ankylosing AND (‘spondylarthritis’/exp OR spondylarthritis)	
#10	Ankylosing AND (‘spondylarthrosis’/exp OR spondylarthrosis)	
#11	‘ankylosis’/exp OR ankylosis AND (‘spondylitis’/exp OR spondylitis)	
#12	Ankylotic AND (‘spondylitis’/exp OR spondylitis)	
#13	Bechterew AND (‘disease’/exp OR disease)	
#14	Bekhterev AND (‘disease’/exp OR disease)	
#15	Morbus AND bechterew	
#16	Spinal AND (‘ankylosis’/exp OR ankylosis)	
#17	‘spine’/exp OR spine AND (‘ankylosis’/exp OR ankylosis)	
#18	‘spondylarthritis’/exp OR spondylarthritis AND ankylopoietica	
#19	‘spondylarthritis’/exp OR spondylarthritis AND ankylosans	
#20	‘spondylarthrosis’/exp OR spondylarthrosis AND ankylopoietica	
#21	‘spondylitis’/exp OR spondylitis AND ankylopoetica	
#22	‘spondylitis’/exp OR spondylitis AND ankylopoietica	
#23	Spondylitis, AND ankylosing	
#24	‘spondyloarthritis’/exp OR spondyloarthritis AND ankylopoietica	
#25	Vertebral and (‘ankylosis’/exp OR ankylosis)	
#26	#4 OR #5 OR #6 OR #7 OR #8 OR #9 OR #10 OR #11 OR #12 OR #13 OR #14 OR #15 OR #16 OR #17 OR #18 OR #19 OR #20 OR #21 OR #22 OR #23 OR #24 OR #25	
#27	‘single nucleotide polymorphism’/exp	
#28	Polymorphism, AND single AND (‘nucleotide’/exp OR nucleotide)	
#29	Single AND (‘nucleotide’/exp OR nucleotide) AND polymorphism	
#30	#27 OR #28 OR #29	
#31	#3 AND #26 AND #30	

Inclusion criteria

Investigations that met the eligibility criteria were included in the analysis, and any disagreements were resolved by discussion between the authors (Xu B and Ma JX). In cases where a consensus could not be reached, a third author was involved to make a final decision. Studies meeting the following criteria were included in the meta-analysis: (1) investigation evaluating the association between IL-23R rs7517847 or rs2201841 polymorphisms and AS susceptibility; (2) a case-control study or GWAS; (3) sufficiently available public data that could be extracted for further analysis, such as genotype distribution, odds ratio (OR) and 95% confidence interval (95% CI); and (4) a SNP distribution according to Hardy-Weinberg equilibrium (HWE) was included, which means high quality in design and conduct of genetic association studies. In cases where two studies examined the same or overlapping populations, the study with a larger sample size was included in the analysis.

Studies for which contact with the main author could not be reached to supply information about relevant data were excluded.

Data extraction and quality assessment

The relevant characteristics of the included investigations were identified and recorded by two authors, including the first author of the study, publication year, country, ethnicity of subjects, relevant SNPs, patient demographics, test method used for genotype, whether the genotype distribution was in accordance with Hardy-Weinberg Equilibrium (HWE) and source of the samples tested. To acquire precise results for the current study, email contact with the main author of the investigation was performed if the included study did not contain public data. Quality assessment of studies included in the meta-analysis was conducted by two authors using the Newcastle-Ottawa Scale (NOS) (Cota et al., 2013). Scores were given for subject selection (i.e., adequateness of the case definition, representativeness of the cases, selection of controls, and definition of controls) and the comparability of the groups (i.e., comparability of cases and controls on the basis of the design or analysis) as well as measurement of exposure (i.e., ascertainment of exposure, same method of ascertainment for cases and controls, and non-response rate). NOS scores ranged from 0 to 9. Studies with a NOS score ≥6 were considered to be high quality. Higgins I2 was used to evaluate the heterogeneity of the investigations. Subgroup analysis by ethnicity was executed if the number of investigations in each ethnic group was two or more. Gender subgroup analysis could not be done because there was no relevant data. Sensitivity analysis was carried out by evaluating the overall results of meta-analysis when each study was removed to detect the stability of trials included.

Statistical analysis

The strength of the association between IL-23R rs7517847 and rs2201841 polymorphisms and AS susceptibility was evaluated using the Odds Ratio (OR). At the same time, precision was measured by 95% CI. A random-effects model was used in the current study. With respect to rs7517847 and rs2201841, the homozygote model, heterozygote model, recessive model, dominant model and allelic model were used to estimate susceptibility to AS in current study. Statistical analysis of the extracted data was carried out using STATA 12.0.

Results

Search results

As a result of the search strategy presented in Fig. 1, 215 English-language studies and 208 Chinese-language studies were obtained. In addition, one study was added by searching through references (Zhu, Yang & Gao, 2009). Among the search results, 420 trials were excluded for the following reasons: (1) 116 were duplicate articles; (2) 281 did not discuss the association between IL-23R SNPs and AS in the title and abstract; (3) 20 did not discuss the association between rs7517847 and rs2201841 and AS in the full-text; (4) two contained genotype distribution data that were not publicly available, and email contact with the main authors could not be achieved; and (5) two studies assessed overlapping populations (Sáfrány et al., 2009; Szabo et al., 2013). For these studies, those that included a larger sample population were included in the analysis, as conducted by Sáfrány et al. (2009). Finally, four trials (Sáfrány et al., 2009; Dong et al., 2013; Zhu, Yang & Gao, 2009; Rueda et al., 2008) on rs7517847 and two trials (Sáfrány et al., 2009; Zhu, Yang & Gao, 2009) on rs2201841 were included in the current study. The included studies had case-control designs with 1,006 patients in the AS group and 1,190 people in the control group for rs7517847. For rs2201841, 322 patients were included in the AS group, and 255 people were included in the control group in the current study. The basic characteristics of included studies were listed in Table 2. Patients who were diagnosed with AS according to the New York modified criteria (Goei et al., 1985) and healthy people from Chinese, Hungarian and Spanish populations were included in the current study. All trials included in this meta-analysis were replication studies according to HWE. The NOS score of studies included in the meta-analysis ranged from 6 to 7, as presented in Table 3. Moreover, genotype distribution of both SNPs in the AS and control groups is presented in Table 4.

Figure 1 PRISMA flowchart of number of studies.

Table 2 Main characteristics of studies included in this meta-analysis.

Author	Year	Country	Ethnicity	SNP	AS group	Control group	Test method	HWE	Source	
Dong H	2013	China	Asian	rs7517847	291	312	PCR-RFLP	Y	Blood	
Zhu XQ	2009	China	Asian	rs7517847	144	143	PCR-HRM	Y	Blood	
Zhu XQ	2009	China	Asian	rs2201841	116	102	PCR-HRM	Y	Blood	
Sáfrány E	2009	Hungary	Caucasian	rs7517847 rs2201841	206	235	PCR-RFLP	Y	Blood	
Rueda B	2008	Spanish	Caucasian	rs7517847	365	500	Taqman	Y	Blood	
Notes.

Author first author

HWE Hardy-Weinberg Equilibrium

Table 3 Quality assessment of studies.

Study included	Selection	Comparability	Exposure	Total	
1. Dong et al. (2013)	3	2	2	7	
2. Sáfrány et al. (2009)	3	2	2	7	
3. Zhu, Yang & Gao (2009)	3	1	2	6	
4. Rueda et al. (2008)	3	1	2	6	

Table 4 Genotype distribution of rs7517847 and rs2201841 associated with IL-23.

			rs7517847		rs2201841		
Author	Ethnicity	Group	TT	GT	GG	MAF	CC	CT	TT	MAF	
Dong H	Asian	AS	104	146	41	0.39	–	–	–	–	
		control	98	153	61	0.44	–	–	–	–	
Sáfrány E	Caucasian	AS	67	115	24	0.40	26	89	91	0.34	
		control	69	126	40	0.44	15	102	118	0.28	
Zhu XQ	Asian	AS	61	59	24	0.37	64	40	12	0.28	
		control	48	77	18	0.40	54	41	7	0.27	
Rueda B	Caucasian	AS	140	172	53	0.38	–	–	–	–	
		control	182	238	80	0.40	–	–	–	–	
Notes.

– no available data

MAF minor allelic frequencies

Overall results of the meta-analysis

ORs, P value, 95% CI and heterogeneity evaluation of meta-analysis for rs7517847 and rs2201841 are presented in Table 5 and Fig. 2 and in Table 6 and Fig. 3, respectively. The random effect model was applied to the studies. A comprehensive analysis of relevant SNPs was performed. Based on the meta-analysis results, there was a statistically significant association between the IL-23R rs7517847 polymorphism in the over-all population and AS susceptibility under two genetic models (homozygote model, i.e., GG vs. TT, were OR =0.76, 95% CI [0.59–0.99] and P = 0.038 and the allelic model, G vs. T, i.e., where OR = 0.88, 95% CI [0.78–0.99] and P = 0.032). The heterogeneity among studies under the two genetic models was I2 = 0, PH = 0.547 and I2 = 0, PH = 0.838, respectively, which indicated that there was no statistically significant heterogeneity among the trials for rs7517847. The analytical results presented in Table 6 implied that there was no statistically significant association between the IL-23R rs2201841 polymorphism and AS susceptibility under all of the genetic models tested.

Figure 2 Forest plot of genetic association studies of rs7517847.

(A) genotype GG vs. TT; (B) allele G vs. allele T. The horizontal axis: axis of Odds Ratio (OR). The dotted line: mean value of overall OR.

Figure 3 Forest plot of genetic association studies of rs2201841 homozygote model (CC vs. TT).

The horizontal axis: axis of Odds Ratio (OR). The dotted line: mean value of overall OR.

Table 5 Meta-analysis of polymorphisms of rs7517847.

SNP	n	Genetic model	OR (95% CI)	P	I 2	
rs7517847	4	Homozygote model GG vs. TT	0.76 (0.59, 0.99)	0.038	0%	
		Heterozygote model GT vs. TT	0.87 (0.73, 1.05)	0.157	0%	
		Allelic model G vs. T	0.88 (0.78, 0.99)	0.032	0%	
		Recessive model GG vs. (TT + GT)	0.82 (0.65, 1.03)	0.090	28.2%	
		Dominant model (GG + GT) vs. TT	0.85 (0.71, 1.01)	0.066	0%	
Notes.

n number of studies included

Table 6 Meta-analysis of polymorphisms of rs2201841.

SNP	n	Genetic model	OR (95% CI)	P	I 2	
rs2201841	2	Homozygote model CC vs. TT	0.87 (0.50, 1.51)	0.631	0%	
		Heterozygote model CT vs. TT	0.93 (0.62, 1.40)	0.726	4.70%	
		Allelic model C vs. T	0.99 (0.77, 1.27)	0.915	0%	
		Recessive model CC vs. (TT + CT)	1.04 (0.69, 1.55)	0.866	0%	
		Dominant model (CC + CT) vs. TT	0.94 (0.64, 1.38)	0.746	0%	
Notes.

n number of studies included

Results of subgroup analysis by ethnicity

A subgroup analysis by ethnicity (Asians and Caucasians) was conducted for rs7517847, and contained two trials (one with 435 and 455 people in the case and controls groups and a second with 571 and 735 people in the case and control group, respectively). The random effect model was applied to each genetic model in both the Asian and Caucasian populations.

In accordance with the results of the subgroup analysis, there was no statistically significant association between the IL-23R rs7517847 polymorphism and AS in the Asian or Caucasian populations, as presented in Table 7.

Table 7 Results of subgroup analysis of rs7517847 by ethnicity.

		Ethnicity	
		Asian	Caucasian	
SNP rs7517847	Genetic model	OR (95% CI)	P value	OR (95% CI)	P value	
	Homozygote model (GG/TT)	0.74 (0.50–1.10)	0.140	0.78 (0.55–1.09)	0.143	
	Heterozygote model (GT/TT)	0.79 (0.59–1.05)	0.109	0.94 (0.74–1.20)	0.613	
	Recessive model (GG/ GT + TT)	0.84 (0.59–1.20)	0.335	0.80 (0.59–1.09)	0.161	
	Dominant model (GG + GT/TT)	0.78 (0.59–1.02)	0.071	0.90 (0.72–1.13)	0.373	
	Allelic model (G/T)	0.84 (0.70–1.02)	0.080	0.90 (0.77–1.05)	0.181	

Sensitivity analysis

A sensitivity analysis of trials investigating rs7517847 was carried out, aiming to evaluate the stability of the current study. After removing the study conducted by Sáfrány et al. (2009), we found that the OR for GG vs. TT changed from OR = 0.76 (95% CI [0.59–0.99], P = 0.038) to OR = 0.80 (95% CI [0.60–1.06] and P = 0.120) and the OR for G vs. T changed from OR = 0.88 (95% CI [0.78–0.99], P = 0.032) to OR = 0.88 (95% CI [0.77–1.01] and P = 0.078). After removing the study conducted by Dong et al. (2013), we found that the OR for GG vs. TT changed from OR = 0.76 (95% CI [0.59–0.99], P = 0.038) to OR = 0.82 (95% CI [0.60–1.11] and P = 0.203) and the OR for G vs. T changed from OR = 0.88 (95%CI [0.78–0.99], P = 0.032) to OR = 0.90 (95% CI [0.78–1.04] and P = 0.145). Excluding the other two studies, we found that the results of studies had not changed.

Discussion

As a member of the erythropoietin receptor superfamily, IL-23R influences the differentiation of CD4 T lymphocytes to IL-17-producing Th17 lymphocytes. During inflammation, IL-17 plays a destructive role, such as articulation of the cerebral, heart, lung and intestinal tissues. Several recent studies have indicated an association between IL-23R polymorphism and some autoimmune diseases, such as inflammatory bowel disease, multiple sclerosis and psoriasis (Duerr et al., 2006; Cargill et al., 2007; Nunez et al., 2008). Overall, four relevant meta-analyses about IL-23R and AS have been conducted and provided statistical evidence for the association between IL-23R polymorphisms and AS susceptibility (Karaderi et al., 2009; Chen et al., 2012; Duan et al., 2012; Pimentel-Santos et al., 2009). The results of two meta-analyses of large populations indicated that associations between 7 SNPs and 6 SNPs of IL-23R and AS existed in Britain and 7 other countries, respectively (Karaderi et al., 2009; Duan et al., 2012).

However, no associations existed for some SNPs in a Portuguese population (Pimentel-Santos et al., 2009) or five SNPs in an Asian population (Chen et al., 2012), in studies where fewer than 1,000 individuals were examined. Additionally, a number of other investigations have assessed the association between IL-23R rs7517847 and rs2201841 polymorphisms and AS susceptibility. However, the outcomes were not conclusive. To obtain a statistically credible result, four studies were included in this meta-analysis. In total, four trials (1,006 AS patients and 1,190 people in the control group) and two trials (322 AS patients and 255 people in the control group) were included for rs7517847 and rs2201841, respectively.

Although no statistically significant association between IL-23R rs7517847 and AS were identified in each of the included studies, the results of the current study showed that there was a statistically significant association between rs7517847 and AS in the overall population in the homozygote model and the allelic model after the four studies were pooled. We speculated that negative results shown in each study were due to the small sample sizes. Additionally, no evidence was obtained to prove the association between rs7517847 and ethnicity in the subgroup analysis. After excluding either the study conducted by Sáfrány et al. (2009) or that Dong et al. (2013), which were conducted in Hungary or China, respectively, we found that the P value of the remaining studies changed to be not significant. However, the odds ratio of the studies were actually fairly stable. (Sáfrány et al. (2009): as for GG vs. TT, OR =0.80, 95% CI [0.60–1.06] and P = 0.120; with respect to G vs. T, OR = 0.88, 95% CI [0.77–1.01] and P = 0.078 Dong et al. (2013): GG vs. TT: OR = 0.82, 95% CI [0.60–1.11] and P = 0.203; G vs. T: OR = 0.90, 95% CI [0.78–1.04] and P = 0.145). The sensitivity analysis was affected by removing the result of either Sáfrány et al. (2009) or Dong et al. (2013), possibly because the remaining sample size after removal of these studies was not sufficiently large. Thus, removing a study could greatly reduce the sample size and lead to changes in the sensitivity analysis. Additionally, no significant association between rs2201841 and AS susceptibility was found in any of the genetic models tested in the current study. The results obtained in this study suggest that the rs71517847 polymorphism might be a protective factor for AS in the overall population but that the rs2201841 polymorphism might not be associated with AS susceptibility. The author (Xu B) speculated that the negative results regarding rs2201841 could have been due to small sample size.

The current study has some limitations as follows. (1) The population included in this study was relatively small. Hence, type II error might exist and the credibility of the current study’s results may be weak. (2) A subgroup analysis of rs2201841 by ethnicity could not be conducted because the eligible studies in each ethnicity that were included in this study only involved one trial such that the association between IL-23R rs2201841 and AS by ethnicity was unknown. (3) There was only one country from Asia and two countries from Europe involved in the current study, which implies that the results of the current study are not representative. (4) Unfortunately, there is a lack of information about related studies for the black population. Therefore, the results of the current study are not comprehensive. (5) We were unable to include studies that have not been published, which might affect the publication bias.

Conclusion

In conclusion, the present meta-analysis showed that the IL-23R rs7517847 polymorphism may play a protective role in AS. However, no association between the rs2201841 polymorphism and AS susceptibility was found. Moreover, to obtain a credible and comprehensive conclusion to enable rheumatologists and researchers in related fields to comprehend the association between rs7517847 and rs2201841 and AS susceptibility, it is still essential to implement further investigations with a large number of samples from more countries to assess the associations between these two SNPs and AS.

Supplemental Information

Supplemental Information 1 PRISMA checklist

Click here for additional data file.

Additional Information and Declarations

Competing Interests

Author Contributions

The authors declare there are no competing interests.

Bin Xu conceived and designed the experiments, performed the experiments, analyzed the data, wrote the paper, prepared figures and/or tables.

Jian-xiong Ma performed the experiments, analyzed the data.

Xin-long Ma performed the experiments, reviewed drafts of the paper.

Hao-bo Jia, Rui Feng and Li-yan Xu contributed reagents/materials/analysis tools.

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
