# Peer review of "Association between rs7517847 and rs2201841 polymorphisms in IL-23 receptor gene and risk of ankylosing spondylitis: a meta-analysis"

_PeerJ, doi:10.7717/peerj.910_

## Round 0.1 · original submission · Major Revisions

· Academic Editor

Major Revisions

While the question addressed by this manuscript is interesting the paper contains a number of grammatical errors. We urge the authors to completely revise the manuscript such that its language would be at publishable quality. Please address all the concerns raised by the two reviewers. Specifically, it would be most valuable to have the random effect model performed as well, as suggested by reviewer 3.

Reviewer 1 ·

Basic reporting

There are many grammatical errors. Many sentences are written in the past tense which caused some confusion. This article would benefit from major copy editing.

Figure 3 and Figure 4 (Forest plots) could benefit from a description of the horizontal axis as well as an explanation of the dotted line meaning to improve interpretation. In Figure 3, should Quan ZX be Zhu XQ?

The results of the subgroup analysis by ethnicity would be better detailed in a table rather than in the text.

Experimental design

I would like to see an adjustment for gender and ethnicity in the overall meta-analysis. It was mentioned in the introduction that gender is an important risk factor for AS in the introduction, however none of the analysis results looks at what impact gender may have.

Validity of the findings

No comments.

Additional comments

Please explain why you chose those databases to search and left out others such as Pubmed.

What is meant by 'invalid line of 95% confidence interval' (Introduction). Do you mean to say 'borderline non-significant'?

Please explain what you mean by 'studies that didn't show sufficient data information' under 'Materials & Methods'.

Under 'Inclusion criteria', please list the two names of the authors that solved discrepancy via discussion.

Please provide some more information on why the Safrany data was 'unstable' and 'not credible'.

Reviewer 2 ·

Basic reporting

1. This paper has a clear focus and is appropriately organized. However, the English usage needs a complete revision. The sentence construction is awkward and often the meaning is not entirely clear. This is a major concern.

2. The first two paragraphs of the Introduction should be written in the present tense.

Experimental design

The research question is clearly defined and in general the methodology used adequately addresses the question. The PRISMA checklist for meta-analyses was followed.

Minor points
Introduction paragraph 3:
1. "Although the association between two SNPs rs7517847 and rs2201841 and AS was not obvious, it exceeded the invalid line of 95%CI slightly." - Please specify which of the references [12-15] has the information on which this statement is based.
2. The way this paragraph is written makes it sound as if there were previous meta-analyses that considered rs7517847 and rs2201841, yet it goes on to say this is the first such meta-analysis. This is confusing.

Inclusion criteria:
3. I know what a case-control study is, but what is a replication case-control study?
4. Please add a sentence about the reason why adherence to the Hardy–Weinberg principle was important when selecting studies.
5. The first two exclusion criteria in the 2nd paragraph can be omitted as they are already implied by the inclusion criteria.

Data Extraction and Quality Assessment
6. "Studies with a NOS score≥6 were considered as studies of high quality and were included in the current meta-analysis." Was this an inclusion criterion? If so it should be listed in the Inclusion Criteria; otherwise the wording should be changed.
7. Information that is in the Statistical Analysis section need not be repeated here.
8. Add a sentence on how the sensitivity analysis was carried out.

Statistical Analysis
9. While I consider it undesirable to choose between fixed and random effects models based on tests of heterogeneity (see for example http://www.campbellcollaboration.org/artman2/uploads/1/2_Pigott_RandomFixedModels.pdf), the practice is quite common and I found some books on meta-analyses advising this course of action so I cannot say it is wrong.

Validity of the findings

In general appropriate tables and figures were provided. The data on which the conclusions are based are provided except for the ellelic frequencies, which could be added to Table 4. There appears to be a mistake in the sensitivity analysis (see 4. below) which can easily be corrected.

Overall results of the meta-analysis
1. "Besides, subgroup analysis of rs7517847 by ethnicity was carried out."
This sentence should be omitted from this section as it is off-topic.
2. "Nevertheless, analytical results implied that there was not statistically significant
association between IL-23R rs2201841 polymorphism … etc.)"
The word "Nevertheless" should be omitted. It is also not necessary to list all the results here as one can refer the reader to Table 6.

Sensitivity analysis
3. Please add the comparable overall results to make it easy for the reader to compare, i.e. "After removing the study conducted by Safrany et al., we found that the OR for GG vs TT changed from OR=0.76 (95% CI=0.59-0.99, P= 0.038) to OR=0.80 (95% CI=0.60-1.06 and P=0.120) and the OR for G vs T changed from OR=0.88 (95% CI=0.78-0.99, P= 0.032) to OR=0.88 (95% CI=0.77-1.01 and P=0.078)."
4. "Excluding the other three studies successively, we found that results of studies were not changed."
It surprised me somewhat that removing the Dong paper did not cause a change in the results similar to the Safrany paper because the two studies are fairly similar in effect size. So I checked the calculations for GG vs TT and found that removing the Dong paper indeed changed the OR to 0.82, P=0.206. So it is not just the Safrany paper that changes the result. I expect similar results for the G vs T comparison. Please recheck and correct your results.

Discussion
5. "The result of sensitive analysis showed that data of the study conducted by Safrany et al. was unstable and evidence for the result was not credible either."
The discussion needs to be adjusted to mention both the Dong and Safrany studies. In addition the sentence above makes it sound as if the Safrany study was somehow defective although this may just be bad English. If, as I suspect, they meant that the meta-analysis results are unstable, this should be placed in context. The estimates of the odds ratio are actually fairly stable. In the case of the G to T comparison the OR did not change at all. It is the confidence interval and p-value that changed which is hardly surprising since by removing a study one is reducing the sample size greatly.

Tables 5 and 6
6. The tables would be easier to read if all five the comparisons were all presented below each other and not split across the page. Also, a column could be added that denotes the Model, with entries homozygote, heterozygote, recessive, dominant and allelic as appropriate. The p-value for heterogeneity could be omitted if space is needed as they are not very important given that the I2 values are already provided.

Figure 2
7. The figure contains an entry for Quan ZX (2009) which is not one of the four listed studies. This needs correcting.
8. The arrow on the Safrany confidence interval is puzzling as the confidence interval stretches from 0.34 which is visible on the x-axis.

Figure 4
9. Figure 4 adds very little additional information as it is essentially Figure 2 with one of the papers removed. I suggest leaving out Figure 4.

Reviewer 3 ·

Basic reporting

Language and statistics

Experimental design

It is a meta-analysis

Validity of the findings

Questionable due to small number of articles.

Additional comments

Comments on ‘Association between rs7517847 and rs2201841 polymorphisms in IL-23 receptor gene and risk of ankylosing spondylitis: A meta-analysis’.

The authors of this manuscript queried the literature and found 4 papers to use in a meta-analysis evaluating the association between rs7517847 and rs2201841 polymorphisms in IL-23 receptor gene and the risk of ankylosing spondylitis. They performed statistical analysis and reported odds ratios and p-values. The authors used a fixed effect meta analytic model as opposed to a random effect model. The authors justified fixed-effect modeling by the absence of heterogeneity. It is not clear how sensitive the test for heterogeneity is when there are only 4 studies. The authors must either justify or include a reference for appropriateness of using fixed-effect method when there are only 4 studies. Alternatively, they could fit the analogous random effect model and discuss the consistency of inference under the different modelling approaches (i.e. do the fixed and random effect models result in the same inference re: the association between these genetic polymorphisms and the risk of ankylosing spondylitis). There are many statistical analyses and sub-analyses performed on a small sample of studies. Validity of the results are questionable. Subject matter experts are in a better position to judge whether this meta analysis of 4 studies is warranted, or whether better inferences may be obtained by waiting for a larger number of studies to be published and then performing the meta analysis at a later date. The authors English and style of writing is poor and the manuscript needs proper editing before publication is warranted.

---

## Round 0.2 · Minor Revisions

· Academic Editor

Minor Revisions

Although the paper has greatly improved, there are still several languages inadequacies. In addition to the ones mentioned by reviewers could the authors also consider the following comments:

1) Please consider the change of the sentence (Inclusion criteria, line 53).
Investigations that met the following criteria
To:
Investigations that met the eligibility criteria
2) There is a dot at the beginning of a sentence in Results of the subgroup analysis by ethnicity (line 123).
3) Please consider changing (Discussion, line 156) the sentence:
….were identified in the included study…
To
… were identified in each of the included studies.
4) Please consider changing (Discussion, line 161) the sentence:
…., which were investigations in Hungary or China…
To
… which were conducted in Hungary and China, respectively..
5) Please consider changing (Discussion, line 163) the sentence:
…. the odds ratios of the studies are actually fairly stable.
To
… the odds ratios of the studies remain fairly stable.
6) Consider the changing the tense of the verbs in the following sentence (discussion, line 178-182).
Instead of:
(3) There was only one country from Asia and two countries from Europe involved in the current study, which implicated that the results of the current study were not representative. (4) Unfortunately, there was a lack of information about related studies for the black population. Therefore, the results of the current study were not comprehensive.
To:
(3) There was only one country from Asia and two countries from Europe involved in the current study, which implies that the results of the current study are not representative. (4) Unfortunately, there is a lack of information about related studies for the black population. Therefore, the results of the current study are not comprehensive.

Reviewer 1 ·

Basic reporting

The manuscript reads very well and reports an interesting meta-analysis.

Some minor errors:
1) Please put a space between 'hymozygote' and 'allelic' in the abstract.
2) Please consider editing the sentence "Although the association between these two SNPs and AS was not obvious, the 95% confidence interval of the SNPs exceeded 1 slightly" to "The association between these two SNPs and AS was borderline significant only".
3) Line 34: please change 'using' to 'uses'
4) Please consider adding a sentence around lines 74-77 to indicate that gender subgroup analysis could not be done because there was no data. Or add to discussion.
5) Line 77: were more than 6 studies ever included in these meta-analyses? If not, please remove this sentence as it causes confusion.
6) Line 103: please consider adding a brief explanation of the interpretation of the NOS score.
7) Line 139: Please add the word 'the' before 'erythroproietin'
8) Line 140: please change the word 'played' to 'plays a'
9) Line 156: Please add 'a' before 'statistically significant'
10) Line 163: please change 'are actually stable' to 'were actually stable'.
11) Line162: Please define 'was changed' as 'changed to be not significant'
12) Line 167: Please consider changing 'total sample size of the meta-analysis was not sufficiently large' to 'remaining sample size after removal of these studies was not sufficiently large'
13) Line 171: please add the word 'the' before 'overall population'
14) Line 172: please define which 'author' you mean.
15) Line 176: please add 'a' before 'subgroup'.

Experimental design

No comments.

Validity of the findings

No comments.

Reviewer 2 ·

Basic reporting

No comment.

Experimental design

No comment.

Validity of the findings

No comment.

Additional comments

The English usage in the paper is vastly improved and of acceptable quality. The authors adequately addressed all the points I raised in the first review. There are only a few small typograpical issues in this version:
1) AS is used in the abstract without defining it.
2) Don't abbreviate SNP in the abstract.
3) line 32 - the wording is still not clear; perhaps try: "the 95% confidence interval of the odds ratio of the SNPs overlapped 1 only slightly."
4) Tables 5 and 6: the heading "OR (95% CI) " should be added. Table 7 has a better layout for the OR and genetic model info and I recommend it be followed in these two tables too.
5) Add the genetic model (CC vs TT) to the legend of Fig 3.

Reviewer 3 ·

Basic reporting

pass

Experimental design

no comments

Validity of the findings

no comments

Additional comments

no comments

---

## Round 0.3 · accepted · Accept

· Academic Editor

Accept

You have adequately answered to the reviewers comments.